# Lipoxygenase in Wheat: Genetic Control and Impact on Stability of Lutein and Lutein Esters

**DOI:** 10.3390/foods10051149

**Published:** 2021-05-20

**Authors:** Daryl J. Mares, Judy Cheong, Shashi N. Goonetilleke, Diane E. Mather

**Affiliations:** 1School of Agriculture Food and Wine, Waite Campus, The University of Adelaide, Urrbrae, SA 5064, Australia; shashi.goonetilleke@adelaide.edu.au (S.N.G.); diane.mather@adelaide.edu.au (D.E.M.); 2SARDI, Waite Precinct, Urrbrae, SA 5064, Australia; judy.cheong@sa.gov.au

**Keywords:** lipoxygenase genes, Asian noodle colour, genetic loci, molecular markers

## Abstract

Preservation of lutein concentrations in wheat-based end-products during processing is important both for product quality and nutritional value. A key constituent involved in lutein degradation is endogenous lipoxygenase. Lutein and lutein ester concentrations were compared at intervals during storage of noodle sheets prepared from flour of wheat varieties representing a range in lipoxygenase activity, as well as in different mill streams and in different grain tissues. Higher lipoxygenase concentration was associated with an increased loss of free lutein and lutein mono-esters whereas lutein diesters appeared to be more resistant to degradation. Lutein degradation was reduced in the presence of a lipoxygenase inhibitor, when noodle sheets were heated to destroy enzyme activity or when pH was increased. In addition, three populations were used to investigate the genetic control of lipoxygenase. A previously reported mutation of *Lpx-B1.1* was associated with a reduction in activity from high to intermediate whilst a new locus on chromosome 4D was associated with variation between intermediate and near-zero. The gene underlying the 4D locus is a putative lipoxygenase. Stability of lutein could be improved by deployment of the mutations at the 4B and 4D loci and/or by post-harvest storage of grain under conditions that promote esterification.

## 1. Introduction

Lutein, a plant carotenoid, is an important micro-nutrient for humans that can only be obtained from foods or supplements. Lutein is responsible for the yellow color of many fruits, flowers and vegetables and much of the creamy to yellow color of food products prepared from cereals such as bread wheat (*Triticum aestivum* L.) and durum wheat (*Triticum durum*). Maintaining lutein levels during production of food products is highly desirable. One factor associated with loss of lutein during processing is lipoxygenase activity. The purpose of this study was to determine the impact of endogenous lipoxygenase on lutein during processing and to identify genetic loci in bread and durum wheat associated with reduced activity. Asian white salted noodles (WSN) were selected as the test end-product due to their ease of production on a small scale and the importance of color and color stability for noodle manufacturers and consumers. The lutein molecule has a hydroxyl group at each end that can be esterified with fatty acids. As a consequence, lutein in wheat grain is present as mixture of different proportions of free lutein, the lutein mono-esters linoleoyllutein and palmitoyllutein, and the lutein di-esters dilinoleoyllutein, linoleolylpalmitoyllutein and dipalmitoyllutein [1,2,3]. Esterification is catalysed by a GDSL esterase/lipase, a plant xanthophyll acyltransferase (XAT) encoded by a gene located on chromosome 7D of bread wheat [4] but not present in durum wheat. The esterification reaction occurs primarily after harvest in the starchy endosperm and to a lesser extent in the grain coat if grain is stored under warm dry conditions [3,5]. The esters appear to be more stable than free lutein to increasing temperature but there is no information available regarding their relative resistance to degradation by lipoxygenase.

Lutein is prone to degradation during food production by a co-oxidation process that involves the enzyme lipoxygenase (LOX, linoleate:oxygen oxidoreductase; EC 1.13.11.12) [6,7]. Lipoxygenase catalyses the deoxygenation of polyunsaturated fatty acids such as linoleic acid to produce hydroperoxide radicals that can then oxidise lutein, other carotenoids and vitamin E. A strong correlation between lipoxygenase activity of flour and the loss of carotenoids during bread making [8]. Similarly, lipoxygenase activity was found to be the main factor involved in the loss of color during durum pasta processing [9]. In this context, it is worth noting that lipoxygenase, often soybean lipoxygenase, is a common ingredient in bread making where it acts as a bleaching agent and appears to enhance dough mixing tolerance and dough rheological properties. The effect of endogenous and exogenous (soybean) lipoxygenase activity on the color of Asian white salted noodles prepared from two commercial white wheat flours has also been investigated [10]. Lipoxygenase activity in noodles remained constant during 24 h storage at room temperature or 4 °C and whilst no apparent reduction in color could be attributed to endogenous lipoxygenase, whiter noodles were obtained following incorporation of soybean lipoxygenase. Nevertheless, color and color stability are still among the major factors that influence flour miller’s buying decisions and consumer product acceptance. The studies noted above have focused on the impact of lipoxygenase on color which is affected by many variables. There appear to be no reports in which the effects of endogenous lipoxygenase on individual lutein and lutein ester compounds during end-product processing were determined.

A lipoxygenase encoded by the *Lpx-B1* gene located on the short arm of chromosome 4B was shown to be associated with variation in lipoxygenase activity in durum wheat grain [11]. Subsequently, a duplication at the *Lpx-B1* locus and allelic variation for a deletion of the *Lpx-B1.1* copy was reported [12]. The deletion was associated with a significant reduction in lipoxygenase and improved pasta color. A similar investigation of the genetic control of lipoxygenase activity in bread wheat does not appear to have been undertaken. The aims of the study were to determine the effects of variation in lipoxygenase activity on free lutein, lutein mono-esters and lutein di-esters during processing and storage of white salted noodles; to quantify the effect of the *Lpx-B1.1* deletion on lipoxygenase activity in two bread wheat and one Australian durum populations; and to identify genetic loci associated with reduced lipoxygenase activity in bread wheat. The results of this work will inform development of strategies that optimize retention of the important micro-nutrient, lutein, in wheat-based foods.

## 2. Materials and Methods

### 2.1. Plant Material and Chemicals

The plant material used here included the Australian durum wheat varieties Kamilaroi and Yallaroi, the Australian bread wheat varieties Sunco, Cranbrook, Halberd, Axe, Sunvale and ‘Yellow’ Kite, the high yellow pigment bread wheat line Indis [13], a synthetic AUS30666, a synthetic hexaploid wheat line developed at CIMMYT, Mexico, materials developed using some of those lines as parents and a panel of 49 other bread wheat varieties (Appendix A). Seeds of Indis were supplied by R.A.McIntosh, University of Sydney. Seed of AUS30666 and of a Cranbrook/Halberd population were obtained from the Australian Grains Genebank, Horsham, Victoria, Australia. A high lutein line, DM5685*B12, was selected at the University of Adelaide from a Sunco*^2^/Indis backcross population. A Kamilaroi/Yallaroi population of 250 single-seed descent lines and an AUS30666/DM5685*B12 population of 183 single seed descent lines were developed at the University of Adelaide. Populations were grown in a glasshouse at the Waite Campus of the University of Adelaide. All chemicals used in the study were supplied by Sigma-Aldrich Pty Ltd., North Ryde, NSW, Australia.

### 2.2. Flour Samples for Determination of Lutein and Lutein Ester Esterification in Asian Noodle Sheets

Large plots of Indis, Sunco and ‘Yellow’ Kite bread wheat and Kamilaroi durum wheat were grown in a field at Narrabri in north-western New South Wales (30.3324° S, 149.7812° E) and harvested immediately once they reached harvest-ripeness (12% moisture content). The bulk sample of Indis grain was immediately separated into three sub-samples. One sub-sample was immediately transferred to −20 °C, one sub-sample was stored dry at room temperature for 2 months whilst a third sub-sample was stored over silica gel at 40 °C for 2 months. Grain of the other varieties were stored dry at room temperature. Storage at 40 °C was chosen as it provided a means of assuring that conditions would favor the formation of lutein esters within the timeframe of the experimental work without significant degradation of lutein or lutein esters [3]. At the completion of the storage period, the grain was conditioned overnight to a moisture content of 15% fresh weight for bread wheat varieties and 16% for the harder grained durum variety. The grain was milled to flour in a Quadrumat Senior mill (Brabender GMBH and Co. KG, Duisburg, Germany, https://www.brabender.com/en/food/products/mills/determine-flour-quality-quadrumat-senior/, accessed on 12 May 2021). The break and reduction flour streams were combined and mixed thoroughly prior to use. Extraction rates ranged from 72–74%.

### 2.3. Preparation of Noodle Sheets

Noodle sheets were prepared according to methods described previously [14]. A 10 g sample of flour was mixed with 3.5 mL of sodium chloride solution 2% (*w*/*v*) for white salted noodles (WSN) or sodium carbonate solution (2% *w*/*v*) for yellow alkaline noodles (YAN), in a small stainless-steel bowl using a paddle attached to an electric drill press. The mixture was formed into a sheet by passing through a hand operated spaghetti machine twice. The pH of WSN and YAN noodles was 6.0 and 8.2, respectively. Noodle sheets were incubated in resealable plastic bags at room temperature and pairs of noodles were taken at 0 h and after 2, 6, 24 and 48 h, frozen in liquid nitrogen, freeze-dried and reduced to a fine powder with a mortar and pestle.

### 2.4. Determination of Lutein and Lutein Ester Concentration

Extraction and quantification of lutein and lutein esters by RP-HPLC was as described in [3].

### 2.5. Determination of Lipoxygenase Activity

#### 2.5.1. Preparation of Samples

Quadrumat Senior mill flour and Buhler mill flour streams were sub-sampled in triplicate. For Buhler milling, grain samples of bread wheat varieties Sunvale and Axe were conditioned overnight to a moisture content of 15% fresh weight before being milled on a Bühler experimental mill (Bühler Group, Uzwil, Switzerland) system (described in https://wwql.wsu.edu/wheat-was/wheat-was-buhler-txt/, accessed on 12 May 2021) produces 3 break flour streams, 3 reduction flour streams, a pollard stream (rich in germ tissue) and a bran stream (rich in grain coat). The pollard stream can be manually re-sieved to recover more flour. In this study, the streams were kept separate for analysis of lipoxygenase content.

Similarly, triplicate samples of five grains from each line of each of the populations were milled to a fine flour in a ball mill for 7 s (RotoMix^TM^, 3M ESPE, St Paul, MN, USA) [3]. To investigate changes in lipoxygenase activity during grain development, 5 spikes were harvested at 5-day intervals, starting at 10 days after anthesis (DPA), from plants growing in a glasshouse. Five grains were removed from each spike and dissected into embryo, grain coat and starchy endosperm. The respective tissues from the five grains were combined, frozen in liquid nitrogen, freeze-dried and ground to a fine meal in the RotoMix.

#### 2.5.2. Lipoxygenase Assay

Lipoxygenase activity was determined using a modification of the method described in [15] that was adapted to a microplate format by Dr R. Leach at the University of Adelaide. Crude extracts were prepared from 0.1 g of flour mixed with 1 mL of 0.05 M sodium acetate buffer at room temperature, pH 5.5. For the tissues derived from developing grains, the entire sample was used. The mixture of flour and extraction buffer was placed on a rotator at 4 °C for 1 h to fully react followed by centrifugation at maximum speed for 10 min in a microfuge. Aliquots (5 μL) of the supernatant were pipetted into wells of a 96 well microplate previously blocked by incubating in 1% bovine serum albumin (*w*/*v*) in water overnight at 4 °C and diluted to 50 μL with sodium acetate buffer. A stock solution of linoleic acid (LA), 7.5 mM, was prepared from pure LA [16]. A 125 μL aliquot of linoleic acid substrate solution (consisting of 100 μL stock LA, 10 mL of 20 mM DMAB (3-dimethylamino) benzoic acid in 50 mM sodium acetate buffer pH 5.5, and 9.9 mL water) was added to each well and plates that were then incubated for 5 min at room temperature. A 125 μL aliquot of a haemoglobin and MBTH (3-methyl-2-benzothiazolinone hydrazone hydrochloride hydrate) solution that consisted of 400 μL 10 mM MBTH and 400 μL of 5 mg mL^−1^ haemoglobin and 19.2 mL water was then added to each well, the solution mixed, incubated for a further 5 min and the reaction stopped by addition of 50 µL/well 0.05% SDS. Optical density was recorded at 595 nm. Lipoxygenase (LOX) activity of flour samples was quantified by measuring absorption at 595 nm with a microplate reader. Results were calibrated against a known amount of linoleic hydroperoxide (LAHP), produced by reacting stock linoleic acid substrate with soybean lipoxygenase (Sigma). LOX activity was reported as nmol LAHP min^−1^ g^−1^, or in the case of developing grain tissues nmol LAHP min^−1^ tissue^−1^.

### 2.6. Genotyping

DNA was extracted from the youngest fully expanded leaf using the phenol/chloroform method [17,18] and quantified using a NanoDrop 1000 spectrophotometer (Thermo Fischer Scientific, Wilmington, DE, USA). For the Cranbrook/Halberd population, an existing linkage map [19] was used. Both Cranbrook/Halberd and Kamilaroi/Yallaroi populations were genotyped with the *Lpx-B1.1* marker (LOXA-L: 5′- CTGATCGACGTCAACAAC-3′, LOXA-R: 5′-CAGGTACTCGCTCACGTA-3′) [12]. For the AUS30666/ DM5685*B12 population, subsets of lines were genotyped by the Centre for AgriBiosciences (Bundoora, Vic, Australia) using a 90k Infinium SNP chip [20]; (26 lines with high LOX activity and 29 lines with low LOX activity) and by Diversity Arrays Technology (Bruce, ACT, Australia) using DArTseq genotyping-by-sequencing technology (www.diversityarrays.com/dart-application-dartseq, accessed on 12 May 2021; 91 randomly selected lines). Marker-trait associations were investigated by single marker regression using Map Manager (QTXb19) [21]. For a chromosome on which SNPs were found to be associated with LOX activity, KASP assays (Appendix A) were designed and were implemented on an automated SNPLine system (LGC Limited, Teddington, UK). Linkage mapping and simple interval mapping for that chromosome were conducted using Map Manager (QTXb19) [21].

## 3. Results

### 3.1. Association between Stability of Lutein and Lutein Esters in Asian Noodles and Endogenous Lipoxygenase Activity

#### 3.1.1. Changes in Lutein and Lutein Ester Concentrations in Noodle Sheets with Time

Lutein and lutein ester concentrations were determined in white salted noodle (WSN) and yellow alkaline noodle (YAN) sheets at intervals during the 48 h after preparation. For the WSN noodle sheets prepared from the Indis samples, the total lutein concentration was similar (Table 1) for all treatments but the proportions of free lutein and lutein esters differed greatly among treatments. In noodles from grain stored at −20 °C, lutein was predominantly present as free lutein (93.5%). By contrast, in noodles from grain stored at 40 °C, free lutein represented only 6% of the total, with lutein mono-esters and lutein di-esters representing 21% and 73%, respectively. During storage of the Indis noodle sheets, the concentrations of all lutein species declined but the reductions in concentration were substantially greater for free lutein and lutein mono-esters than for lutein diesters (Table 1). A substantial proportion (up to 90%) of the total losses occurred in the first 6 h, whereas further reductions at 24 and 48 h were comparatively small despite there being more than half the original lutein remaining. The percent changes in the two lutein mono-esters, monolineolyllutein and monopalmitoyllutein, were of similar magnitude and for simplicity the concentrations were combined and reported as lutein monoesters. Similarly, concentrations of the three lutein di-esters were combined and reported as lutein di-esters. The greater stability of lutein di-esters relative to free lutein and lutein mono-esters was reflected in the total lutein concentrations in noodles at each of the sampling times for the Indis samples that had been stored under three different temperature regimes; 40 °C > room temperature > −20 °C. Similar patterns of reduction in lutein and lutein ester concentrations were observed for Sunco and ‘Yellow’ Kite noodles. In contrast, the reduction in concentration of free lutein and total lutein in noodles prepared from the high lutein durum wheat variety Kamilaroi was much lower at 8.7% and 12.4%, respectively, at 48 h (compared to 54% and 53% in in Indis noodles derived from the −20 °C-derived sub-sample). A direct comparison between noodles prepared from Indis and ‘Yellow’ Kite, both of which had been stored at room temperature indicated that whilst the proportions of free lutein and lutein esters were similar at 0 h, the reduction in total lutein concentration by 48 h was greater for ‘Yellow’ Kite (51%) than for Indis (40%). When noodle sheets were immersed in boiling water for 3 min, then cooled in cold water, there was little loss of lutein or lutein esters over the ensuing 48 h (data not shown). Yellow alkaline noodles (YAN) were also prepared for the Indis sub-sample that had been stored at room temperature and Kamilaroi. In contrast to WSN, free lutein, lutein mono-esters and total lutein concentrations did not decrease substantially during the first 6 h after preparation. Concentrations of lutein di-esters were in fact higher at all sampling times than at 0 h in Indis noodles (Table 1). Reductions in concentrations of all lutein species were observed between 24 and 48 h such that by 48 h the reduction in total lutein was 12% in Indis and 25% in Kamilaroi.

#### 3.1.2. Lipoxygenase Activity

Lipoxygenase activities in grain and flour of the wheat varieties used to investigate lutein and lutein ester stability were ranked in order ‘Yellow’ Kite > Indis > Sunco > Kamilaroi (Table 2) and appeared to be inversely related to the stability of lutein in WSN. Activity remained constant during the 2-month storage period used in this investigation. When the lipoxygenase assay was run in the presence of 0.1 mM n-propyl gallate, L-ascorbate or α-tocopherol, activity was reduced by 78, 83 and 95% relative to control respectively. These lipoxygenase inhibitors were also incorporated into WSN noodle sheets prepared from ‘Yellow’ Kite and Kamilaroi flour. Over the first 6 h, the percent reduction in total lutein in ‘Yellow’ Kite noodles was 34% (control), 21% (n-propyl gallate), 18% (L-ascorbate) and 7% (α-tocopherol) compared with 7, 5, 4 and 3% respectively for Kamilaroi noodles.

Grains of the high lipoxygenase variety Halberd were sampled at 20 and 30 days after anthesis (DPA) in triplicate and dissected into embryo, starchy endosperm and grain coat tissue, dried and assayed for lipoxygenase activity. Embryo tissue, starchy endosperm and grain coat tissues represented 4%, 44% and 52% of grain dry weight at 20 DPA and 2%, 55% and 42% at 30 DPA. Activity in embryo tissue, measured as nmol min^−1^ g^−1^, was very high relative to activity in starchy endosperm and grain coat (Table 3). However, when activity was calculated per tissue segment, by 30 DPA the total activity in the embryo and grain coat was similar whilst that in the starchy endosperm remained very low.

Samples, replicates of 5 grains, were harvested from AUS30666, Halberd, Cranbrook and Kamilaroi at 5-day intervals after anthesis starting at 10 days and dissected into embryo and non-embryo fractions. Lipoxygenase activity, measured as nmol min^−1^ tissue segment^−1^, started increasing between 15 and 20 DPA in Halberd embryos and reached maximum activity at 40 DPA (Figure 1A). Activity in the embryos of the other varieties remained low throughout grain ripening. In de-embryonated grains, activity increased from between 20 and 25 DPA in Halberd, between 25 and 30 DPA in Cranbrook and remained low in AUS30666 and near zero in Kamilaroi (Figure 1B).

Bulk grain lots of two wheat varieties, Axe and Sunvale, were milled on a Buhler mill and samples of each mill stream collected for assay of lipoxygenase activity. Total flour recovery (break + reduction + sieved pollard) was 77.7% for Sunvale and 77.5% for Axe. The higher grain activity of Axe was reflected in higher activities in the bran and pollard streams (Figure 2). Flour lipoxygenase activities were 6.7 and 9.2 nmol min^−1^ g^−1^ for Sunvale and Axe respectively compared with 51 and 84 nmol min^−1^ g^−1^ for bran + pollard. Overall, the results were consistent with the grain tissue comparison (Table 3) however it was not possible to determine the relative contribution of embryo and grain coat lipoxygenase to the flour (break and reduction streams).

### 3.2. Genetic Control of Lipoxygenase Activity in Bread and Durum Wheat

In the Cranbrook (medium LOX activity)/Halberd (high LOX activity) population, a QTL (LRS = 55.5) for LOX activity was mapped near the *Lpx-B1.1* and *Rht-B1* loci on the short arm of chromosome 4D (Appendix A). Cranbrook, which contributed the low LOX activity allele at the QTL, carries the deletion allele at *Lpx-B1.1* and the dwarfing allele at *Rht-B1*. This QTL explained 32% of the phenotypic variation in the Cranbrook/Halberd population, with lines with and without the *Lpx-B1.1* deletion allele both exhibiting a range of LOX activities (Figure 3A). Genotyping of the Kamilaroi/Yallaroi population for the *Lpx-B1.1* deletion revealed a very strong association of the *Lpx-B1.1* polymorphism with LOX activity. In that population, all lines with the deletion allele had very low LOX activity, while lines without that allele exhibited a range of LOX activities (Figure 3B).

A further population, derived from a cross between AUS30666 (a very tall near-zero LOX synthetic hexaploid wheat) and DM5685*B12 (a semi-dwarf line with medium LOX activity), was investigated in an attempt to target LOX activity lower than the medium levels associated with the best current commercial varieties of bread wheat. Both parents carry the *Lpx-B1.1* deletion. Based on application of genome-wide genotyping to subsets of lines, SNPs in a region of chromosome 4D were found to be significantly associated with variation in LOX activity. KASP assays were designed for 34 SNPs on chromosome 4D and applied to the full population to construct a linkage map for chromosome 4D (Appendix A). QTL analysis for LOX activity data from two years revealed a very highly significant QTL (LRS = 173.1 in one year, accounting for 61% of the phenotypic variation; LRS = 114.1 in the other year, accounting for 46% of the phenotypic variation) (Appendix A). At the closest marker to the QTL peak, an A > C SNP in D_contig37975_58 (at position 16,356,092 on the 4D pseudomolecule of the IWGSC RefSeq Version 21 wheat genome assembly), 120 AUS30666/DM5685*B12 lines exhibited the A:A genotype and only 60 lines exhibited the C:C genotype. All lines with the C:C genotype had low LOX activity, whilst lines with the A:A genotype exhibited a wide range of LOX activity values (Figure 4). When this SNP was assayed on 49 bread wheat varieties (Appendix A), none of them exhibited the C:C genotype, indicating that it could be used as a diagnostic marker for near-zero LOX activity. The D_contig37975_58 SNP is within 0.5 Mb of the *Lpx-D1* lipoxygenase gene (KC679302). Attempts to amplify parts of *Lpx-D1* from AUS30666 were unsuccessful, indicating that AUS30666 may lack this gene.

## 4. Discussion

The results of this investigation are consistent with endogenous lipoxygenase playing a role in the degradation of lutein during the processing of wheat-based end-products such as pan bread, flat bread and white salted noodles. All of these products are prepared using recipes that are based on flour/salt/water mixtures with a pH of around 6, which is close to the optimum pH for wheat lipoxygenase activity [16,22]. Whilst free lutein and lutein mono-esters appeared to be similarly susceptible to degradation, the di-esters were much more resistant. This raises the possibility of reducing degradation by storing wheat grain under warm, dry conditions that promote esterification prior to use [3]. Free lutein, which carries hydroxyl groups at each end of a highly conjugated polyene chain, is highly reactive and susceptible to both oxidative degradation by light and heat and to co-oxidation involving lipoxygenase. In addition to protecting against degradation by lipoxygenases, lutein esterification appears to confer stability at temperatures above 40 °C [3].

Lipoxygenase activity was found to be concentrated in the embryo and seed coat of the grain. Consistent with this, lipoxygenase activity was higher in the germ or pollard (bran) streams from Buhler milling than in flour. Nevertheless, significant lipoxygenase activity was recovered in the flour, due to activity in the starchy endosperm and/or contamination of flour with germ and bran during the milling process.

At the high pH levels used to produce yellow alkaline noodles, wheat grain LOX had little activity, making this product largely insensitive to its presence. LOX activity was also inhibited by heat, as in boiling water or frying in oil, but, in practice, noodle dough sheets are left to rest for some hours prior to cutting into noodles strips for cooking or sale as fresh product, leaving ample time for lutein degradation by lipoxygenase. Lipoxygenase activity can also be inhibited by α-tocopherol (vitamin E), a strong antioxidant that is sometimes incorporated into noodle sheet recipes. Whilst wheat grains contain significant amounts of endogenous α-tocopherol, it is concentrated in the embryo with only relatively low levels or very low levels in bran and flour respectively [23].

Ultimately, genetics is likely to provide a more effective means of stabilizing the lutein concentration in wheat-based products. The *Lpx-B1.1* deletion previously shown to be associated with reduced loss of colour in durum pasta [12] was shown in this study to substantially reduce the level of lipoxygenase activity in bread wheat. In the population used here, the *Lpx-B1.1* deletion allele was linked in coupling phase with the dwarfing allele at *Rht-B1* locus. The linkage between these two important loci may need to be taken into account in wheat breeding if selection is to be imposed for the preferred alleles at one or both loci. In contrast to durum wheat, a survey of bread wheat varieties failed to identify any lines with near-zero lipoxygenase activity [24]. However, a small number of synthetic hexaploid wheat lines were identified and one of these was used to create the population, AUS30666/DM5685*B12, that was exploited in this study. Since both parents carried the *Lpx-B1.1* deletion allele, this allowed the variation in lipoxygenase in the range intermediate to near-zero to be targeted. A highly significant locus on chromosome 4D explained a high proportion of this variation. It seems likely that the cause of this QTL is a deletion of or in *Lpx-D1*. A closely linked SNP was identified that could be used in marker-assisted selection.

## Figures and Tables

**Figure 1 foods-10-01149-f001:**
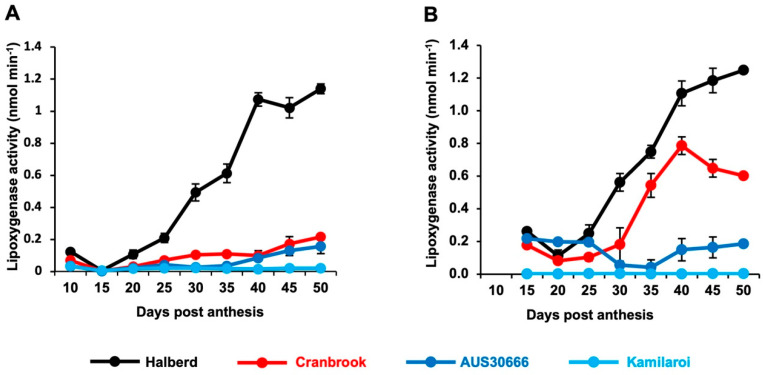
Lipoxygenase activity in (**A**) embryo tissue and (**B**) de-embryonated grain tissue dissected from grains at intervals after anthesis. Lipoxygenase activity is expressed as nmol min^−1^ tissue^−1^. Bars represent standard errors.

**Figure 2 foods-10-01149-f002:**
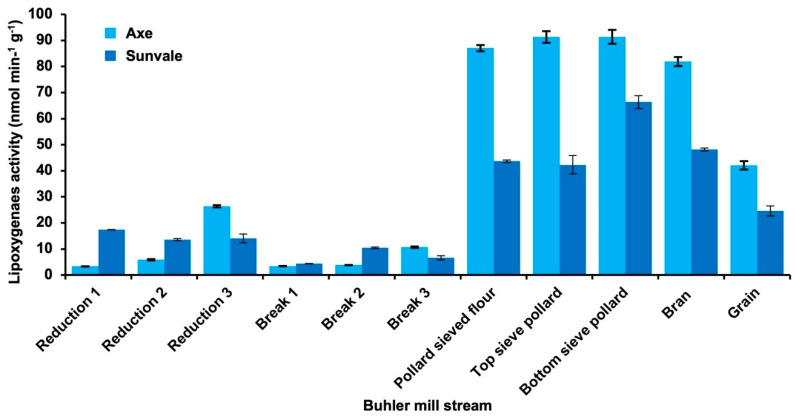
Lipoxygenase (LOX) activity in Buhler mill streams compared with the activity in the grain of two Australian wheat varieties.

**Figure 3 foods-10-01149-f003:**
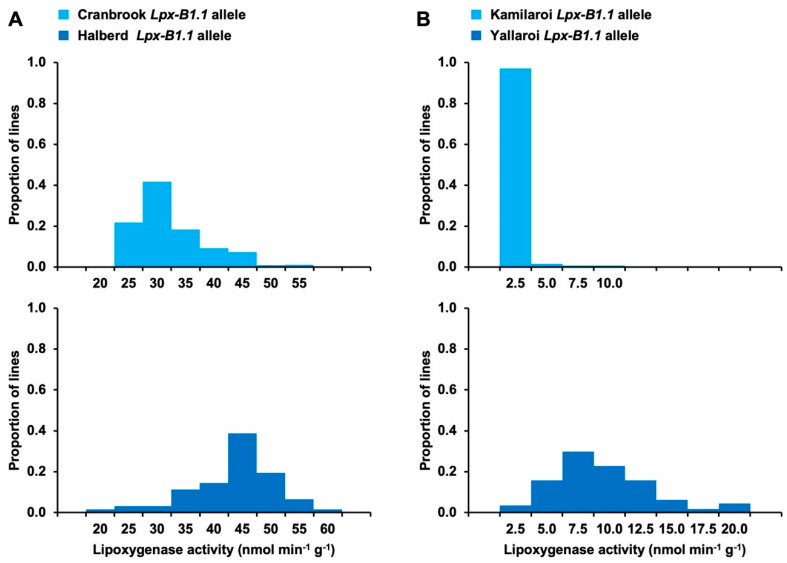
Lipoxygenase activity in (**A**) Cranbrook/Halberd and (**B**) Kamilaroi/Yallaroi. Upper panels show the phenotypic frequency distributions for lines that are homozygous for the *Lpx-B1.1* deletion allele from (**A**) Cranbrook and (**B**) Kamilaroi. Lower panels show the phenotypic frequency distributions for lines that are homozygous for the *Lpx-B1.1* alleles from (**A**) Halberd and (**B**) Yallaroi.

**Figure 4 foods-10-01149-f004:**
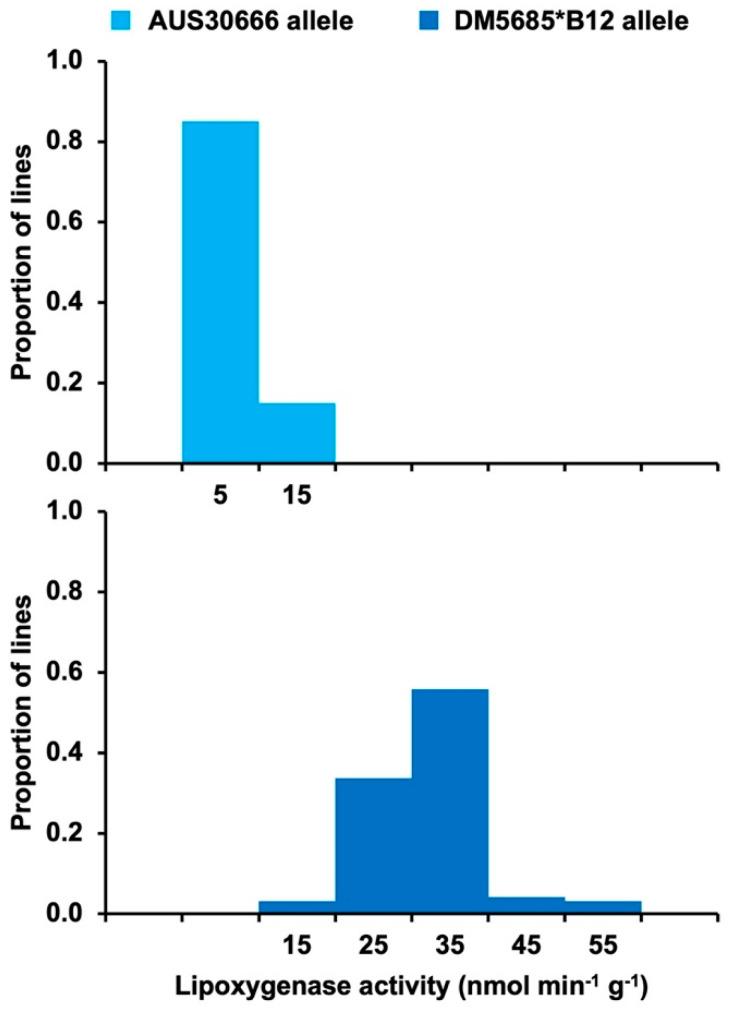
Lipoxygenase activity (based on means across two years) in AUS30666/DM5685*B12. The upper and lower panels show phenotypic frequency distributions for lines that are homozygous for the alleles from AUS30666 and DM5685*B12, respectively, at the QTL on chromosome 4D.

**Table 1 foods-10-01149-t001:** Lutein and lutein ester concentrations in (**a**) white salted noodle (WSN) and (**b**) yellow alkaline noodle (YAN) sheets sampled 2, 6, 24 and 48 h after preparation. Figures after the ± symbol are standard deviations.

a. WSN	Concentration (μg Lutein Equivalents g^−1^)				
	0 h	2 h	6 h	24 h	48 h
Indis (Stored at −20 °C from harvest)					
Lutein	5.94 ± 0.02	4.68 ± 0.05	3.07 ± 0.05	2.85 ± 0.02	2.74 ± 0.01
Mono-esters	0.35 ± 0.01	0.16 ± 0.01	0.13 ± 0.01	0.12 ± 0.002	0.12 ± 0.001
Di-esters	0.05 ± 0.01	0.05 ± 0.01	0.05 ± 0.01	0.04 ± 0.01	0.05 ± 0.01
Total	6.34 ± 0.003	4.89 ± 0.07	3.25 ± 0.03	3.01 ± 0.006	2.91 ± 0.002
Indis (Stored at room temperature for 2 months)					
Lutein	3.63 ± 0.01	2.97 ± 0.006	2.08 ± 0.06	2.06 ± 0.02	1.91 ± 0.01
Mono-esters	0.65 ± 0.01	0.53 ± 0.008	0.33 ± 0.01	0.30 ± 0.01	0.30 ± 0.01
Di-esters	1.89 ± 0.02	1.80 ± 0.01	1.52 ± 0.05	1.58 ± 0.015	1.53 ± 0.02
Total	6.17 ± 0.04	5.30 ± 0.02	3.93 ± 0.08	3.94 ± 0.03	3.74 ± 0.01
Indis (Stored at 40 °C for 2 months)					
Lutein	0.37 ± 0.01	0.32 ± 0.01	0.24 ± 0.01	0.23 ± 0.005	0.21 ± 0.006
Mono-esters	1.46 ± 0.01	1.23 ± 0.004	0.87 ± 0.01	0.83 ± 0.02	0.82 ± 0.04
Di-esters	4.36 ± 0.03	4.06 ± 0.15	3.91 ± 0.09	3.91 ± 0.07	3.90 ± 0.08
Total	6.19 ± 0.03	5.61 ± 0.14	5.02 ± 0.11	4.97 ± 0.08	4.93 ± 0.10
Sunco					
Lutein	0.78 ± 0.02	0.62 ± 0.04	0.52 ± 0.004	0.52 0 ± 01	0.46 ± 0.003
Mono-esters	0.16 ± 0.01	0.12 ± 0.015	0.10 ± 0.003	0.09 ± 0.01	0.08 ± 0.004
Di-esters	0.34 ± 0.01	0.33 ± 0.08	0.31 ± 0.003	0.31 ± 0.015	0.29 ± 0.01
Total	1.28 ± 0.002	1.07 ± 0.07	0.93 ± 0.001	0.92 ± 0.01	0.83 ± 0.003
Kamilaroi					
Lutein	6.98 ± 0.29	7.25 ± 0.002	6.69 ± 0.002	6.26 ± 0.01	6.15 ± 0.05
Mono-esters	1.39 ± 0.11	1.48 ± 0.01	1.26 ± 0.01	1.13 ± 0.02	1.11 ± 0.016
Di-esters	0	0	0	0	0
Total	8.37 ± 0.17	8.73 ± 0.01	7.95 ± 0.01	7.39 ± 0.02	7.26 ± 0.07
‘Yellow’ Kite					
Lutein	2.00 ± 0.02	1.59 ± 0.04	0.92 ± 0.01	0.92 ± 0.05	0.88 ± 0.05
Mono-esters	2.13 ± 0.01	1.69 ± 0.01	0.98 ± 0.006	0.98 ± 0.01	0.14 ± 0.004
Di-esters	0.55 ± 0.006	0.50 ± 0.01	0.35 ± 0.02	0.34 ± 0.02	0.57 ± 0.015
Total	4.68 ± 0.02	3.78 ± 0.03	2.25 ± 0.006	2.24 ± 0.025	1.59 ± 0.03
**b. YAN**					
Indis (Stored at room temperature for 2 months)					
Lutein	3.99 ± 0.04	4.08 ± 0.004	3.98 ± 0.01	3.82 ± 0.02	3.09 ± 0.01
Mono-esters	0.57 ± 0.003	0.66 ± 0.015	0.58 ± 0.01	0.56 ± 0.01	0.51 ± 0.01
Di-esters	1.33 ± 0.055	1.94 ± 0.013	1.96 ± 0.01	1.89 ± 0.02	1.62 ± 0.02
Total	5.89 ± 0.02	6.68 ± 0.003	6.52 ± 0.01	6.27 ± 0.05	5.22 ± 0.04
Kamilaroi					
Lutein	7.81 ± 0.02	7.93 ± 0.02	7.71 ± 0.01	6.62 ± 0.005	5.82 ± 0.02
Mono-esters	1.64 ± 0.02	1.60 ± 0.03	1.56 ± 0.03	1.30 ± 0.06	1.21 ± 0.01
Di-esters	0	0	0	0	0
Total	9.45 ± 0.04	9.53 ± 0.05	9.27 ± 0.04	7.92 ± 0.05	7.03 ± 0.014

**Table 2 foods-10-01149-t002:** Lipoxygenase activity in grain and flour of wheat varieties. nd = not determined.

Variety	Lipoxygenase Activity (nmol min^−1^ g^−1^)	
	Grain	Flour
Indis	29 ± 2.2	3.8 ± 0.04
Sunco	18 ± 1.5	1.6 ± 0.03
Kamilaroi	0.49 ± 0.08	0.05 ± 0.003
‘Yellow’ Kite	48 ± 2.6	5.5 ± 0.02
DM5685*B12	25 ± 2.8	nd
AUS30666	1 ± 0.2	nd
Cranbrook	22 ± 3	nd
Halberd	53 ± 2.5	nd

**Table 3 foods-10-01149-t003:** Lipoxygenase activity in embryo, starchy endosperm and grain coat of Halberd sampled at 20 and 30 days. Activity was calculated both as per g dry weight and per tissue segment.

Days after Anthesis		Tissue Dry Weight	Lipoxygenase Activity	
	Tissue Type	(g)	(nmol min^−1^ g^−1^)	(nmol min^−1^ Tissue Segment g^−1^)
20	Embryo	0.005 ± 0.0004	470 ± 17	0.42 ± 0.005
	Endosperm	0.055 ± 0.014	1.1 ± 0.1	0.01 ± 0.0001
	Grain coat	0.064 ± 0.006	3.9 ± 0.1	0.05 ± 0.0002
30	Embryo	0.004 ± 0.0004	153 ± 30	0.32 ± 0.004
	Endosperm	0.11 ± 0.01	1.5 ± 0.2	0.03 ± 0.0002
	Grain coat	0.08 ± 0.012	20.5 ± 1.5	0.40 ± 0.003

## Data Availability

Not applicable.

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
