# Peer review of "Lipoxygenase in Wheat: Genetic Control and Impact on Stability of Lutein and Lutein Esters"

_foods, 2021, doi:10.3390/foods10051149_

Round 1

Reviewer 1 Report

A very good paper. but needs minor revision

  1. Lines 106-108: Suggest to provide the QJ milling method and extraction rates of the flour samples.
  2. Lines 124-125: Provide the method for Buhler test mill and how the flour streams were collected and analyzed.
  3. How the durum samples were milled?
  4. While free lutein was mostly converted into lutein diesters during wheat storage at 40oC for two months, the detrimental impact on gluten properties and shortened shelf life far outweighs its benefits in protecting the loss of lutein from LOX induced oxidation.

Author Response

Lines 106-108: Suggest to provide the QJ milling method and extraction rates of the flour samples.

Response: the following text was inserted….15% fresh weight for the bread wheat varieties and 16% for the harder-grained durum variety. The grain was milled to flour in a Quadrumat Senior mill (Brabender GMBH & Co, KG, Germany, https://www.brabender.com/en/food/products/mills/determine-flour-quality-quadrumat-senior/). The break and reduction flour streams were combined and mixed thoroughly prior to use. Extraction rates ranged from 72 -74%.

Lines 124-125: Provide the method for Buhler test mill and how the flour streams were collected and analyzed.

Response: The following text was inserted. For Buhler milling, grain samples of bread wheat varieties Sunvale and Axe were conditioned to 15%. The Buhler mill (Bühler Group, Switzerland) system (described in: https://wwql.wsu.edu/wheat-was/wheat-was-buhler-txt/) produces 3 break flour streams, 3 reduction flour streams, a pollard stream (rich in germ tissue) and a bran stream (rich in grain coat). The pollard stream can be manually re-sieved to recover more flour. In this study, the streams were kept separate for analysis of lipoxygenase content.

How the durum samples were milled? See response to point 1. The durum was treated the same as the bread wheat except that grain was conditioned to 16% rather than 15% to allow for greater grain hardness of durum.

While free lutein was mostly converted into lutein diesters during wheat storage at 40oC for two months, the detrimental impact on gluten properties and shortened shelf life far outweighs its benefits in protecting the loss of lutein from LOX induced oxidation.

Response: the following text was inserted. Storage at 40°C was chosen as it provided a means of assuring that conditions would favour the formation of lutein esters within the timeframe of the experimental work without significant degradation of lutein or lutein esters [3].

Note: Ahmad et al. 2013 [3] reported that the rate of esterification increased with increasing temperature, however at temperatures greater than 40°C there was an increasing loss of total lutein. We are not advocating that millers should store grain at

40°C although this may be unavoidable in some maritime climates.

Reviewer 2 Report

Congratulations to the Authors for a significant research paper about Lipoxygenase activity and noodle quality. Specifically, genetic markers 4D and 4B will be useful for marker-assisted breeding wheat varieties with stable noodle color. Interesting practical application of high-temperature noodle storage and stability of lutein di-esters of importance from processors point of view. I would like to suggest authors expand this study using wheat varieties of global importance and representing global diversity.

Author Response

An interesting suggestion but one which we are no longer in a position to address.